# Identification, Phylogenetic and Expression Analyses of the *AAAP* Gene Family in *Liriodendron chinense* Reveal Their Putative Functions in Response to Organ and Multiple Abiotic Stresses

**DOI:** 10.3390/ijms23094765

**Published:** 2022-04-26

**Authors:** Lingfeng Hu, Ruifang Fan, Pengkai Wang, Zhaodong Hao, Dingjie Yang, Ye Lu, Jisen Shi, Jinhui Chen

**Affiliations:** 1Co-Innovation Center for Sustainable Forestry in Southern China, Key Laboratory of Forest Genetics and Biotechnology of Ministry of Education, Nanjing Forestry University, Nanjing 210037, China; hlf625@njfu.edu.cn (L.H.); a1530791977@163.com (R.F.); pkwang@szai.edu.cn (P.W.); haozd@njfu.edu.cn (Z.H.); yangdj@njfu.edu.cn (D.Y.); luye@njfu.edu.cn (Y.L.); jshi@njfu.edu.cn (J.S.); 2College of Horticulture Technology, Suzhou Polytechnic Institute of Agriculture, Suzhou 215000, China

**Keywords:** *Liriodendron chinense*, *AAAP*, systematic evolution, expression pattern, abiotic stress

## Abstract

In this study, 52 *AAAP* genes were identified in the *L. chinense* genome and divided into eight subgroups based on phylogenetic relationships, gene structure, and conserved motif. A total of 48 *LcAAAP* genes were located on the 14 chromosomes, and the remaining four genes were mapped in the contigs. Multispecies phylogenetic tree and codon usage bias analysis show that the *LcAAAP* gene family is closer to the *AAAP* of *Amborella trichopoda*, indicating that the *LcAAAP* gene family is relatively primitive in angiosperms. Gene duplication events revealed six pairs of segmental duplications and one pair of tandem duplications, in which many paralogous genes diverged in function before monocotyledonous and dicotyledonous plants differentiation and were strongly purification selected. Gene expression pattern analysis showed that the *LcAAAP* gene plays a certain role in the development of *Liriodendron* nectary and somatic embryogenesis. Low temperature, drought, and heat stresses may activate some *WRKY/MYB* transcription factors to positively regulate the expression of *LcAAAP* genes to achieve long-distance transport of amino acids in plants to resist the unfavorable external environment. In addition, the *GAT* and *PorT* subgroups could involve gamma-aminobutyric acid (GABA) transport under aluminum poisoning. These findings could lay a solid foundation for further study of the biological role of *LcAAAP* and improvement of the stress resistance of *Liriodendron*.

## 1. Introduction

As a result of natural selection, there are currently only two species of *Liriodendron* in nature, namely, *L. chinense* and *L. tulipifera*, which are distributed in Asia and North America, respectively. Both are widely grown in China, mainly for landscaping and timber supply [1]. However, in the process of planting *Liriodendron*, adverse external environmental conditions such as temperature extremes and the presence of heavy metals ions will still affect the growth of *Liriodendron*, which hinders its popularization. Therefore, understanding the growth, development, and resistance mechanism of *Liriodendron* is conducive to its promotion and application.

Amino acids, as the main circulation form of organic nitrogen in the process of plant growth and development, are involved in various life processes, including protein synthesis, hormone regulation, and energy storage, as well as nucleotides, chlorophyll, some plant hormones, and most secondary metabolites [2,3]. It is a necessary source of nutrition and an important regulatory mechanism. The absorption of amino acids in the substrate by plants is achieved mainly by acid transporter proteins (*AAT*s), thereby completing the transmembrane transport of amino acids through the phloem and xylem of plants, thereby completing the transmembrane transport of amino acids through the phloem and xylem of plants in plants [4,5]. The *AAT* superfamily was divided into two families, amino acid/auxin permease (*AAAP*) and amino acid polyamine choline (APC) gene families [6]. Among them, the *AAAP* gene family includes eight subgroups: amino acid permeases (AAPs), lysine histidine transporters (LHTs), proline transporters (ProTs), γ-aminobutyric acid transporters (GATs), putative auxin transporters (*AUX*s), similar to ANT1-like aromatic and neutral amino acid transporters (ANTs), and amino acid transporter-like (ATLa and ATLb) [6]. The protein structures of different subfamilies of *AAAP*s are very different. It is generally believed that the diversification of the sequences and structures of *AAAP*s is due to the specificity of the amino acids corresponding to amino acid transporters [7]. As a class of transmembrane transporters, the conserved domain of *AAAP* contains multiple transmembrane domains, the number and position of which are relatively conserved in different proteins and species; and the changes of transmembrane domains in transporters also indicate that genes are involved in transport functional differentiation [8].

*AAAP* protein is mainly involved in regulating the long-distance transport of amino acids in the body in plants, mediating amino acid transport across membrane structures, and participating in a variety of other life processes [9]. So far, *AAAP* genes in several species have been identified, such as *Arabidopsis* [10], rice [11], moso bamboo [12], *Fragaria vesca* [13], and *Brassica rapa* [7] et al. The function of many *AAAP* genes verified in model species. In *Arabidopsis thaliana* and *Oryza sativa*, *AtAAP1* mediates the amino acid transport in embryo and root cells [14,15]; *AtAAP2* could impact the metabolism of seed yield and oil content through the xylem-phloem transfer [16]; *AtAAP3* is exclusively expressed in the root involved in amino acid uptake from soil [8] and OsAAP3 could control the yield by influencing the amino acid transfer [17]; *AtAAP4* also plays a role in phloem loading [9]; AtAAP5 may perform amino acid import in companion cells in different organs [18,19] and *OsAAAP5*. The amino acid regulates tiller number in rice [20]; *AtAAP6* could uptake amino acid function in the xylem parenchyma [21,22] and overexpression of the *OsAAP6* cold enhance root absorption and improve the nutritional quality in rice [23]. Inhibition of *StAAP1* expression results in altered amino acid levels in the phloem of *Solanum tuberosum* [16]. Overexpression of *GmAAP6a* enhanced soybean nitrogen tolerance and source-sink transport capacity and improved soybean seed quality [24]. *AtAAP8* plays a crucial role in the early seed development in *Arabidopsis* [25]. *PtAAP11* mediates proline transport with high affinity, providing proline to cell wall proteins during xylem formation in *Populus trichocarpa* [26]. Other subgroups have also been researched comprehensively. Members of the *LHT* subfamily are specific for lysine and histidine. *AtLHT1*, *2*, *4*, *5*, and *6* are thought to play an important role in plant sexual reproduction and are expressed in male and female floral organs [9,27]. In addition, *AtLHT1*, *4*, and *6* were detected in the root and bud, respectively [11,27,28]. *OsLHT6* is specifically expressed in new shoot meristems [27]. *PgLHT* is involved in the growth and development of *Panax ginseng* roots [29]. The *AUX* gene family mainly promotes the development of roots or shoots by maintaining the homeostasis of auxin inside and outside cells together with other auxin-related proteins [30,31,32]. In the *Gossypium hirsutum*, *GhAux1*, *GhAux2*, *GhAux3*, *GhAux6*, and *GhAux7* are mainly expressed in the vegetative organs of cotton and participate in the vegetative growth of cotton. However, *GhAux4* and *GhAux5* are preferentially expressed in the ovule of cotton on the day of flowering, and are mainly involved in the initiation of cotton fibers [33]. As a specific transporter of GABA, the expression of *AtGAT1* is increased under flower and GABA treatment [34]. The ProTs subfamily is responsible for the transport of proline, glycine betaine (GB) and GABA in the vegetative organs of roots, leaves, and phloem and phloem in barley [35,36]. Overexpression of *GmProT1* and *2* can affect the synthesis and response of proline and alleviate the damage of salt and drought stress to plants [37]. *GsGAT* gene expression levels increase the ability of Glu to convert to GABA and transport GABA to maintain high nitrogen availability in *Camellia sinensis* [38].

Although many functional *AAAP* genes have been revealed, proving their importance in the process of plant growth and development and in resistance to abiotic stress [36,37], the *AAAP* gene in *L. chinense* has not been characterized. In order to explore the composition of *AAAP* gene family members in *L. chinense* and the possible biological processes involved, we identified and analyzed them based on the whole genome data of *L. chinense* and systematically analyzed the *AAAP* gene family from the aspects of phylogeny, gene function differentiation, expression pattern, and stress response, which has provided the foundation for further gene function research and genetic improvement.

## 2. Results

### 2.1. Identification of LcAAAP Gene Family

Through the local BLASTP and HMMER to search the *AAAP* protein in the *L. chinense* genomic database, 73 putative protein sequences were obtained. Next, the Pfam and Conserved Domain Databases were applied to verify the conserved domains. Finally, the 52 *LcAAAP* protein sequences were retained; these genes were designated based on the At*AAAP* gene family (Table 1, Figure 1 and Appendix A). Gene characteristics, including chromosomal location, open reading frame length, amino acid length, molecular weight, isoelectric point, and subcellular location were analyzed (Table 1). Through analysis of the chromosome distribution based on the *L. chinense* genome, 48 *LcAAAP* genes were located on 14 chromosomes. Chromosome 4 had the most *LcAAAP* genes (14), followed by chromosome 2 (seven genes), chromosomes 1 and 17 (five genes), chromosome 14 (four genes), chromosomes 3 and 18 (three genes), chromosomes 11, 13, and 16 (two genes), and chromosome 8, 10 and 12 (one gene). The genes LcAPP6c, *LcATL7d*, LcATL7b, and LcATL8a were located on the contigs. The *LcAAAP* proteins ranged from 288 (LcAPP17) to 1109 (LcAPP7a) amino acids, with molecular weights between 31.31 (LcAPP17) and 123.62 (LcAPP7a) kDa. The predicted pI values ranged from 5.56 (LcATL8b) to 9.9 (*LcPorT1*). The subcellular location predicted result showed the 48 genes can be located in the cell membrane (Table 1). LcATL1b only was located on the chloroplast, and LcAPP6a, LcATL8a, and LcAPP6b only were located on the Golgi apparatus. Additionally, the 17 genes could be located in other parts of the cell without the cell membrane, such as the chloroplast, cytoplasm, mitochondrion, and nucleus. There was a certain relationship between multiple positioning sites and gene function.

### 2.2. Evolutionary Analysis of LcAAAP Genes

To explore the *LcAAAP* gene family’s evolutionary relationship with other species, a phylogenetic tree was constructed based on the *AAAP* protein sequences of *L. chinense* (Lc), *Oryza sativa* (Os), *Arabidopsis thaliana* (At), *Amborella trichopoda* (Atr), *Sorghum bicolor* (Sb), *Populus trichocarpa* (Pt), *Zea mays* (Zm), and *Vitis vinifera* (Vv) (Appendix A). The basic information on this protein is listed in Appendix A. The *AAAP* gene family could be divided into eight subgroups: Lysine and Histidine Transporters (LHT), Amino acid permease (AAP), proline transporter (ProT), gamma-aminobutyric acid transporter (GAT), auxin transporter (*AUX*), Amino acid-like transporter (ATLa and ATLb), and Aromatic and neutral amino acid transporters (ANT). The *LcAAAP* gene family also was divided into eight subgroups, including 6 LHT members, 15 AAP members, 2 ProT members, 5 GAT members, 5 *AUX* members, and 2 ANT members. A total of 10 members were classed into ATLa, and seven members belong to the ATLb (Figure 2). The phylogenetic tree result showed that many members of *LcAAAP* were clustered into a clade with *VvAAAP* and *AtrAAAP*; it was indicated that the *AAAP* gene family of *L. chinense*, *A. trichopoda* and *V. vinifera* have a closer kinship (Figure 1 and Appendix A).

Understanding gene structure can provide information about evolution and gene function. Therefore, the gene structures, protein conserved motif., and transmembrane topology were analyzed. The *LcAAAP* contains different exon numbers from 2 to 15 (Figure 3). Different subfamilies show different patterns of gene structure. Most *AAAP* gene family members within each subgroup share the same or similar gene structure and gene length. However, there were also certain differences in the structure of genes between members of the same subfamily: for example, *LHT4* only has two exons, but 15 exons were detected in *LHT2*. Gene structure analysis showed that the gene structure of the *LcAAAP* gene family was relatively diverse, which may indicate the evolutionary trend of diversity and diverse gene functions. At the same time, if the positions of exons and introns in the same subfamily are relatively conserved, this may indicate that members of the same subfamily share a close evolutionary relationship. In addition, we found that multiple *LcAAAP* gene family members possess ultra-long introns (>10 kb); this result may negatively affect gene transcription. Motif analysis indicated that each subfamily has its own unique motif distribution pattern (Figure 4 and Appendix A). Most of the genes that were revealed to be closely related by the phylogenetic analysis had a conserve motif type and distribution. Most of the genes that were revealed to be closely related by the phylogenetic analysis had a conserved motif type and distribution. Moreover, some conserved motifs were widely distributed among *AAAP* members, such as motif1, 4, and 6. In contrast, most conserved motifs are specific, appearing only in specific subfamilies. The motifs 14,18,19, and 20 belong only to the *AUX* subgroup, and motif 5 only exists in the APP subgroup; this result may herald their association with functions unique to subfamilies. In addition, some motifs such as 9,11,12,13, and 17 were mainly distributed in one subgroup, but they were sporadically distributed among individual members of other subfamilies; this may affect changes in the function of certain genes.

Most *AAAP* gene family members are responsible for the transport of amino acids and mediate their transport across the cell’s membrane structure. This indicates that the transmembrane structure of the AAAP protein has a specific connection with the function of the gene. We predicted the *LcAAAP* gene family transmembrane domain (Figure 5 and Appendix A and Appendix A). The result showed that the *LcAAAP* gene family typically has 8–12 transmembrane domains, and the location and number of transmembrane domains were relatively conserved, indicating that members of the same subfamily may have similar functions. However, there were also some members whose transmembrane domains were quite different, such as *LcATL1**a* (14), *LcLHT2* (17), *LcAPP4**b* (17), *LcATL11* (4), *LcAAP17* (5), *LcATL9* (6), and *LcPorT1* (6). Reduced or expanded numbers of transmembrane domains indicate that their functions may change.

### 2.3. Synteny Analysis of LcAAAP Genes in L. chinense, Grape, Arabidopsis, and Rice

The expansion and contraction of gene families are affected mainly by whole genomic duplication and by tandem and segmental duplication. *L. chinense* has a single lineage-specific WGD event that occurred approximately 116 million years ago. This may cause the expansion of the *LcAAAP* gene family members. In addition, the whole-genome collinearity analysis result showed that one pair of tandemly duplicated even members (*LcGAT1a-LcGAT1c*) and six pairs of segmentally duplicated members (*LcAPP4a*-*LcAPP4b*, *LcAUX1a-LcAUX1b*, *LcLHT1-LcLHT2*, *LcLAX5a-LcLAX5b*, *LcATL5a-LcATL5c*, *LcLAX2-LcLAX5b*) were detected in the *LcAAAP* gene family (Figure 6, Appendix A). The *GAT*, *APP*, *AUX*, *ATL*, and *LHT* subgroups have member expansion through tandem and segmental duplication, especially the *AUX* subgroups. Moreover, the number of collinear genes shared by different species may reflect the gene family phylogenetic relationships. We performed the synteny analysis of the *AAAP* gene family between *L. chinense*, *Arabidopsis thaliana*, *Oryza sativa*, and *Vitis vinifera* (Figure 7). A total of 38 orthologous pairs were investigated between *L. chinense* and the other three species’ *AAAP* genes. Among them, 10, 4, and 24 orthologous pairs were presented with genes in *A. thaliana*, rice, and grape, respectively. The *LcAAAP* and *VvAAAP* gene families have a large number collinear gene pairs, which shows that *LcAAAP* and *VvAAAP* are closer in evolutionary position.

To verify whether the duplicated homologous genes of *AAAP* were selected during the evolution to adapt to external changes, we measured the Ka/Ks nucleotide substitution ratios of collinear genes to study the exerted selective pressure (Appendix A). In the *LcAAAP*, we found that their Ka/Ks << 1, indicating that the duplicated *LcAAAP* genes have undergone a purifying selection during their evolutionary history. Compared with the *Arabidopsis thaliana*, *Oryza sativa*, and *Vitis vinifera*, most of the *AAAP* gene pairs Ka/Ks << 1, it was mean that the orthologous *LcAAAP* genes of the three species were subjected to purifying selection during evolution. However, for two gene pairs between *LcAAAP* and *VvAAAP* gene families Ka/Ks>1, it was indicated that the two pairs have positive selection during their evolution.

### 2.4. Codon Usage Bias Analysis

The codon usage pattern reflects the evolution and mutation of species or genes. The relative synonymous codon usage and the relative frequency of synonymous codons of *LcAAAP* genes were calculated with CodonW (Appendix A). Here, 30, 33, and 1 codon had a positive bias, negative bias (RSCU < 1), and no bias (RSCU =1), respectively. In the more frequently used codons, some amino acids (RSCU > 1) such as Leucine (Leu), Isoleucine (Ile), serine (Ser), and threonine (Thr) have two or more codons with positive bias. Moreover, if an RFSC value exceeds 60% or is 0.5 times greater than the average frequency of synonymous codons, the codons are considered high frequency. The RFSC analysis result showed that UUC, AAG, and AGC showed high frequency. Compared with the *At**AAAP*, *Os**AAAP*, and *AtrAAAP* codon usage frequency ratios, 1, 29, and 1 codon had ratios greater than 2.00 or lower than 0.50, respectively. Among them, 29 codons had significant differences with the *AAAP* genes of *O. sativa*. This result showed that the *LcAAAP* codons usage frequency is like that of *AtAA**AP* and *AtrAAAP* and that Arabidopsis can be used as one of the choices for *LcAAAP* gene receptors.

### 2.5. Analysis of Cis-Acting Elements in LcAAAP Promoters

To further investigate which process could be regulated through the *LcAAAP*, the cis-acting element promoter was analyzed by the online tool PlantCARE. We detected 111 type cis-acting elements and multiple elements that could be involved in different processes. Among them, 43 type elements were related to light responsiveness within all the *LcAAAP* genes (Appendix A). In addition, we mainly analyzed and classified cis-acting elements related to hormones and stress. (Figure 8). Here, 11 and 3 type elements related to hormones and stress were identified, respectively. Most obviously, the ABRE (cis-acting element involved in the abscisic acid responsiveness) elements were abundantly enriched in certain genes, such as *LcATL7d/8b/9* and *LcAVT6* et al. Furthermore, the elements related to drought induction (MBS), low temperature responsiveness (LTR), auxin response (AAuxRR-core and TGA-element), and MeJA response (CGTCA-motif and TGACG-motif) were also recognized in many *LcAAAP* promoters. It was suggested that these genes may be involved in hormone signaling and stress response. In general, different gene promoters have different regulatory elements, suggesting the complexity of gene regulatory networks. In Figure 6, we found that 51 *LcAAAP* genes have one or more stress response elements. Among them, 32 and 37 genes could respond to low temperature signals and drought signals by MYB gene binding sites, and 18 genes could be involved in defense and stress responsiveness. Auxin, abscisic acid, salicylic acid, gibberellin, and methyl-jasmone response elements were distributed on all *LcAAAP* gene promoters. The most widely distributed was the ABA element (50), followed by the MeJA element (44 genes), GA response element (40), auxin response element (33), and SA response element (1).

### 2.6. Expression Patterns of LcAAAP Genes in Various Organs, Somatic Embryogenesis, and Response to Different Stress

We combined the published transcriptome data to determine the expression patterns of individual *LcAAAP* genes in seven organs (bract, leaf, shoot apex, stamen, petal, pistil, and sepal) (Figure 9). A total of 52 *LcAAAP* genes were divided into four groups based on their expression profiles in a hierarchical clustering heat map. In group 1, the 13 *LcAAAP* genes (*LcAUX1a/1b*, *LcLHT1/4/8/11*, *LcATL5a/7b/9*, *LcAPP2/7b*, *LcLAX5a*, and *LcAVT6*) had a high transcripts per million (TPM) value in various organs, except for the LcLHT8 expression level in the leaf. It was suggested this group gene could have an important role in the flower. In group 2, the genes expression level of the six *LcAAAP* genes (*LcATL1b/5c*, *LcAPP7a/7c*, *LcANT2*, and *LcPorT2*) maintain a steady state in different organs. The remaining 7 genes (*LcAPP1a/4a* and *LcATL1a/7b/8b/12a/12b*) were only specifically expressed in certain organs. In groups three and four, 12 genes (*LcAPP1b/4b/4c/6b/6c/6d/17*, *LcGAT1b/1c/4*, *LcLHT3*, and *LcPorT1*) lacked expression or had low expression levels. The L*cATL15* and *LcLHT2* were specifically highly expressed in the stamen; it was suggested that they could be involved in stamen development. The *LcLAX5b* only was found in the shoot apex. In addition, some genes only were detected at a relatively high expression level in the leaf, shoot apex, stamen, and pistil, such as *LcLAX5b* and LcLHT6, which had higher expression levels only in the shoot, apex, and pistil.

We compared the expression levels of the *LcAAAP* gene family in mid-petal development (1–4) of *L. chinense* and *L. tulipifera* (Figure 9 and Appendix A). The expressions of some members of the second and fourth groups caught our attention. In *L. tulipifera*, the expressions of *LcAPP1a/1b/4a* and *LcANT8a* were upregulated to a very high level (58-fold, 67-fold, 79-fold, and 23-fold, respectively) in the second or third stage of petal development, whereas in *L. chinense* petals, the expression levels of these genes were low or significantly reduced. In contrast, the expression level of LcLHT8, *LcATL9*, and Lc*AUX*1a in the petals of *L. tulipifera* was higher than that of *L. chinense* through the development of petals. In addition, we also explored the expression pattern of *LcAAAP* during somatic embryogenesis of hybrid *Liriodendron* (Figure 9 and Appendix A). Based on the laboratory-established hybrid *Liriodendron* somatic embryogenesis system, we divided the process into 11 stages (PEM: embryogenic callus; ES1: 10 days after liquid culture; ES2: 2 days after screening; ES3: ABA 1 day of treatment; ES4: ABA treatment for 3 days; ES5: globular embryo; ES6: heart-shaped embryo; ES7: torpedo embryo; ES8: immature cotyledon embryo; ES9: mature cotyledon embryo; PL: plantlet). The expression levels of *LcLHT1* and *LcATL5a* were continually upregulated 23.9-fold and 22.5-fold in the embryogenic callus stage (ES1–ES4), respectively. In *LcLHT1*, *LcATL5a*, and *LcAUX1a/1b*, there were certain genes (*LcLHT1/8*, *LcATL1a*/5a/8, *LcAUX1a/1b*, *LcAPP4/9*, and *LcLAX2/5a*) that were consistently highly expressed in the somatic embryogenesis stage, and the *LcAUX1a/1b* were stably expressed in all 11 stages of hybrid *Liriodendron* somatic embryogenesis. These results indicate that these genes may be involved in some important regulatory pathways in the process of somatic embryogenesis of hybrid *Liriodendron*.

We determined the expression pattern of the *LcAAAP* gene family under drought, low temperature (4 °C), and heat stress in the leaf (Figure 10). The 52 members were clustered into multiple groups following the different stresses. In response to the heat (40 °C) treatment, four groups were found by gene expression level. The group 3 gene (*LcAPP4a*, *LcLHT1*, and *LcATL12b*) expression levels were downregulated quickly in a short period of time (12 h) and then upregulated. In group 2, the *LcATL5a* and *LcATL9* gene expression levels showed a trend of downregulation and upregulation within 12 h after heat stress treatment, respectively, and then recovered. Downregulated expression levels were detected after 1h of treatment for six genes (*AUX1a/1b*, *LcAPP1a/2/7b*, *LcLAX5b*). In the other genes of groups 1 and 4, the expression levels of most genes in leaves were very low or not expressed. Under drought stress, we did not observe significant up- or downregulation of *LcAAAP* gene expression except for *LcATL5a*, which was downregulated as processing time grew. In addition, the expression levels of some members only changed at a certain point, and eventually all returned to the original pattern, such as the *LcAPP4a* and the *LcAAP7b*, which upregulated only at 24 h and 12 h, respectively. Regarding the effect of low temperature on gene expression, *LcATL5a*, *LcAPP4a*, and *LcLHT1* upregulated after 12 h exposure to 4 °C. The *LcAPP7b* and *LcAUX1a/1b* showed a trend of downregulated expression. In addition, we found that the *LcATL12b* expression level increased rapidly in 12 h. In this result, the expression abundances of most genes changed in a short period of time during different stress treatments, but the changes were small. It was worth noting that *LcATL5a*, *LcAPP4a*, and *LcLHT1* showed different expression patterns under multiple stress treatments. These results suggest that diverse mechanisms control *LcAAAP* gene responses to various stresses.

### 2.7. Prediction and Correlation Analysis of LcAAAP Interacting Proteins

The construction of protein interaction networks is of great significance for the study of gene interactions and regulatory relationships. We demonstrate the protein interaction between the *LcAAAP* gene family and the protein interaction between *LcAAAP* protein and *LcMYB/WRKY* transcription factor (Figure 11 and Appendix A). The protein interaction network within the *LcAAAP* gene family was divided into three groups. The first group, *LcANT2*, can interact with *LcAPP4c/6c/9* and *LcGAT2/4*, and *LcPorT1* can interact with *LcATL1b/8a* as the second group. In addition, a complex network of interactions involving some of the other *LcAAAP* members was constructed in the third group. *LcLAX2*, *LcAPP2*, and *LcATL9* can interact with each other. *LcATL7d* can interact with the *LcLAX2*, *LcAPP2*, *LcATL12b*, and LcANT1. Some transcriptional regulators that may be involved in the regulation of *LcAAAP* protein were also identified (Figure 11). The result showed that LcMYB118 and LcWRKY22 could interact with *LcAPP2*, and *LcMYB40/108* could interact with *LcLAX2*, respectively. *LcAUX1b* can be independently regulated by multiple transcription factors (*LcMYB77/89/108/186*), and *LcWRKY18*, *LcMYB91*, and *LcMYB25* can be regulated with *LcAPP7b*, *LcAVT6*, and Lc*LHT4*, respectively.

To further explore whether *LcAAAP* was involved in heat, drought, and low temperature stresses (Figure 12 and Appendix A), we analyzed the correlation between *LcAAAP*s and the *LcMYB/WRKY* transcription factors involved in abiotic stress. The result showed a positive correlation (r > 0.7) between the expression levels of *LcAPP4a* and *LcWRKY28*, *LcMYB83*, *LcMYB141*, *LcWRKY5*, *LcMYB71*, and *LcMYB38*. *LcLHT1* showed a positive correlation with *LcWRKY14*, *LcWRKY15*, *LcWRKY16*, *LcWRKY17*, *LcWRKY27*, *LcMYB9*, *LcMYB37*, *LcMYB38*, *LcMYB83*, and *LcMYB105*. In the same way, there was a significant positive correlation among the expression levels of *LcATL12b*, LcMYB28, *LcMYB38*, *LcMYB56*, and *LcWRKY17*. These results further showed that LcAPP4, *LcLHT1*, and *LcATL12b* could be involved in the abiotic stress response process. In addition, some genes may be specifically regulated only in certain abiotic stresses: for instance, under heat stress, the expression level of *LcAPP2* has a positive correlation with *LcMYB32*, *LcMYB43*, *LcMYB84*, *LcMYB89*, *LcMYB110*, *LcMYB113*, *LcMYB138*, and *LcMYB144*. A negative correlation (r < −0.7) was identified in *LcMYB12* and *LcMYB84*. In summary, the correlation analysis shows that a few *LcAPPPs* participate in the stress response.

### 2.8. Response of GAT and PorT Subgroups to Al Stress

Wang et al. showed that gamma-aminobutyric acid (GABA) signaling could enhance tolerance of hybrid *Liriodendron* to Al stress by promoting organic acid transport and maintaining cellular redox and osmotic balance. The current study shows that GAT is specific for GABA transport, whereas PorT mainly transports proline, but PorT of some species can also respond to GABA. Therefore, we explored the responses of GAT and PorT family members under the three treatment methods: Al stress, GABA treatment (Figure 13, Figure 14 and Figure 15), and Al stress followed by GABA treatment. In this process, the six genes of the GAT and PorT subgroups have varying degrees of expression except for LcGAT1b, which were not detected.

In the Al stress, the *LcGAT1a*, *LcGAT1c*, *LcGAT2*, and *LcGAT4* increase the expression abundance in root and stem after 48 h under Al stress. *LcGAT2* was upregulated in leaves after 48 h treatment, while *LcGAT4* expression in leaves and shoots began to increase after 24 h treatment. *LcPorT1* was consistently upregulated in different organs. LcPorT2 showed two expression patterns in different organs, downregulated first and then upregulated in stem and bud, in contrast to expression in leaves.

Under GABA stress, the expression level of *LcGAT1a* in roots and shoots was downregulated after 24 h of treatment, and the expression of *LcGAT1c* in leaves and shoots was downregulated after 12 h of treatment. Compared with other *LcGAT* genes, the expression level of *LcGAT2* in the stem continued to increase after 12 h of treatment. In the PorT subgroup, *LcPorT1* could be upregulated in four organs after 12 h of stress. Interestingly, the expression trend of LcPorT2 in stems, leaves, and shoots was the inverse of that of *LcPorT1*, suggesting that their responses to GABA may have mutually antagonistic effects.

*LcGAT1a* and *LcPorT1* were upregulated in stem and leaf at 12 h after the GABA was applied after the Al stress treatment. A similar situation exists in other GAT genes. For example, expression levels of *LcGAT1a*, *LcGAT1c*, and *LcGAT2* were upregulated in leaves at 12 h or 48 h; *LcGAT2* also was upregulated in stem and bud at 12 h and 24 h, respectively. In addition, the downregulated abundance genes compared with the Al stress in different organs also were detected. *LcGAT1a*, *LcGAT1c*, *LcGAT2*, and *LcPorT1* were downregulated in root, stem, and bud after 48 h of treatment. LcPorT2 was downregulated in stem and bud at 48 h.

The above analysis results show that the hybrid *Liriodendron* may respond to changes in the external environment by changing the content of specific substrates in cells of different organs, thereby affecting the expression of the *LcGAT* and LcPorT subgroups’ genes.

## 3. Discussion

As one of the largest gene superfamilies of *AAT*, the *AAAP* gene family can be divided into eight subfamilies according to existing research, which plays an important role in plant physiology and stress resistance research and has great potential value. Relevant research work has been carried out in a variety of model plants. such as *Arabidopsis*, rice, potato, and polar et al. [10,11,39,40]. Although the functions and roles of *AAAP* genes have been explored in multiple species, systematic analysis of the *AAAP* gene family has not been completed after the publication of the genome of *L. chinense*. In this research, 52 *LcAAAP* genes were identified based on the genome of *L. chinense*, and eight subgroups were divided by the *AAAP* gene family phylogenetic tree of multiple species. Although the cluster clade was like othe*r AAAP* gene families, the numbers of members in different subgroups differs significantly, suggesting that different degrees of expansion occurred in different species.

In addition to *LcAPP6c*, *LcATL7d*, *LcATL7b*, and *LcATL8a*, the remaining 48 genes were unevenly mapped to 14 chromosomes, and most genes were distributed in chromosomes 4, 2, and 17. Subcellular localization results showed that 48 *LcAAAP* genes were located on the cell membrane; it was suggested that these genes were mainly used for membrane localization to experiment with the transport of substances inside and outside the cell. Among the *LcAAAP* genes, many genes have multiple locations (17) or are located only in the chloroplast or Golgi (4). It was indicated that these genes could also function in different organelles. To better understand *LcAAAP* gene function and evolutionary history, we explored different perspectives. Exon–intron structure analysis indicated that most *LcAAAP* genes in the same subgroup had the same or similar numbers. However, there are still structural inconsistencies in some genes, such as differences in the number of exons and the existence of ultra-long introns. Most *AAAP* gene family members are responsible for the transport of amino acids with different specificities and characteristics and mediate their transport across the cell’s membrane structure. It was suggested that the existence of the transmembrane (TM) domain plays an extremely important role in the *AAAP* genes. 7–11 TM were identified in most of the *LcAAAP* genes of the same subgroups. However, there are still some genes *(LcLHT2/4*, *LcATL11*, *LcPorT1*, and *LcAPP4/17*) whose numbers of TMs have expanded or shrunk. Compared with the results of the motif analysis, the conserved motif types of different subfamilies were found to be consistent with the transmembrane domains. This result indicates that the composition of the transmembrane domain is closely related to gene function and further suggests a direct connection between gene motifs and gene function.

WGD and gene duplication events were thought to be the main causes of gene family expansion and functional diversity [41,42]. In the Magnolia plants, two and one whole-genome duplication events occurred in *C. kanehirae* and *L. chinense* [1,43], which could lead to the members of the gene family expanding. Currently, 25% (13/52) of *AAAP* genes are duplicated genes in *L. chinense*, which exist as one pair of tandemly duplicated and six pairs of segmental duplicated events, especially in the *LcAUX* subgroup, including three pairs of gene duplication events. These results showed that the segmentally duplicated event was the main cause of expression of the *LcAAAP* gene family. In addition, we analyzed the collinear relationship of *AAAP* genes between *L. chinense*, *Arabidopsis thaliana*, *Oryza sativa*, and *Vitis vinifera*. The 24 orthologous gene pairs were identified between *LcAAAPs* and *VvAAAPs*, greater than the 10 and 4 orthologous gene pairs found in *LcAAAP-AtAAAP* and *LcAAAP-OsAAAP*. It was indicated that there may be a closer kinship between *LcAAAPs* and *VvAAAPs.* At the same time, there were collinear relationships between the *LcAAAP* genes and *VvAAAP/AtAAAP/OsAAAP*, which also proves that the *LcAAAP* genes may have been formed before monocotyledonous and dicotyledonous plants. These opinions can be verified from a genome-wide perspective: magnoliids arose before the divergence of eudicots and monocots, represented by *Liriodendron* [1]. The Ka/Ks ratio of orthologous genes’ duplicated events was used to evaluate the *AAAP* genes contribute to organism fitness: Ka/Ks < 1 for most paralog gene pairs, both within and between species, which indicated that these paralog gene pairs encountered various degree purifying selective pressures, demonstrating that complex selective pressures drove the evolution of the *AAAP* gene family. Gene duplication events provide raw materials for regulating physiological and morphological changes in plants and provide a basis for species to adapt to changes in the external environment [44]. *LcAUX1a* and *LcAUX1b* have similar expression patterns in various organs and stresses, which indicated that they may have redundant functions. However, the other paralog gene pairs all showed different expression patterns, which, because of the duplicated genes, always tend to be sub-functionalized, neo-functionalized, or both [45]. This result suggested that the strong purifying selection may have inflected the gene functional divergence to adapt to diversity in the environment.

There was a very important relationship between the biological function of genes and their expression level, and a comprehensive analysis of the expression level of the genes can analyze the putative gene function in the process of plant growth and development [46]. In this study, the expression profiles of the *LcAAAP* gene family were analyzed in different organs and stages of *Liriodendron*. Approximately 28 and 34 *LcAAAP g*enes were expressed at relatively high levels (TPM > 1) in the leaves and shoot apex of *L. chinense*, respectively. Here, 40 genes showed relatively high expression levels in the flowers of *L. chinense*. We found that *LcLHT1*, *LcLHT2*, and *LcATL15* have high expression in leaves, bract, and stamens, respectively. This result is like that of paralogous genes in Arabidopsis [19,28,47]. Therefore, *LcLHT2* could play an important role in amino acid transport in pollen development and maturation, and *LcALT15* also might be involved in the long-distance transport of amino acids in stamens. In addition, the expression levels of *LcLHT8* were significantly higher in floral organs than in leaves; it was indicated that *LHT8* could also have an important role in flowering. Nectar generally contains substances such as carbohydrates, amino acids, inorganic ions, proteins, and lipids [48] and the central area of the petals was thought to be the location of the *Liriodendron* nectary [49,50]. The *LcAAP1a* gradually upregulated during the development of nectar in *L. chinense*; and the expression levels of *LcAAP1a*, *LcAAP1b*, and *LcAAP4* in the nectar of *L. tulipifera* were significantly upregulated in the second stage relative to *L. chinense*, which indicated that the increased levels of amino acid transport in the nectar implies that the amount of nectar secreted by *L. tulipifera* would be greater than that of *L. chinense*, which is consistent with reality. This result showed that a more complex amino acid transport network exists in the nectar secretion of *L. tulipifera*. As protein synthesis substrates, amino acids play an indispensable role in the process of plant somatic embryogenesis [51,52]. As the embryogenic cells develop toward the globular embryo stage in hybrid *Liriodendron* (PEM-ES4), the *LcLHT1* and *LcATL5a* were specifically upregulated in this progress. During subsequent embryonic development (ES5-ES8), the *LcLHT1/8*, *LcATL1a/5a*, and *LcAAP4a* maintained stable levels, and the expression levels of *LcAUX1a*, and *LcLAX2* were significantly upregulated relative to the previous stage. In addition, the *LcATL5a* and *LcAAP9* were upregulated, and the *LcLAX2* and *LcLAX5a* were downregulated during plant morphogenesis (ES9-PL). These results indicate that protein and auxin transport plays an important role during somatic embryogenesis. In the first stage, a large amount of protein accumulates as the basis for subsequent embryonic development. In the second stage, the content of intracellular auxin decreases, resulting in assists in embryonic development, similar to the expression pattern of *LcPIN* genes [53]. In the third stage, the specific regulation of auxins and proteins in cells provides the basis for plant morphogenesis.

According to previous studies, *AAAPs* were involved in the low temperature, heat temperature, drought stress treatments in many plants [35,54,55]. In this study, we analyze the *LcAAAP* gene expression pattern based on the stress transcriptome data of hybrid *Liriodendron* [56,57]. A total of 13 genes in hybrid *Liriodendron* leaves were able to respond to the three abiotic stress treatments. Most genes’ expression levels were downregulated in a short period of time. Interestingly, the significant changes occurred in the expression patterns of *LcATL5a*, *LcAPP4a*, and *LcLHT1* in these data, and they showed completely opposite trends in heat and low temperature stress treatments. Cis-acting element analysis results showed that the promoters of *LcAAAP* genes’ response to the abiotic have at least one of the LTR (response to the low temperature), MBS (MYB binding site involved in drought-inducibility), and TC-rich repeat elements (cis-acting element involved in defense and stress responsiveness), which showed that promoters could control structural and morphological changes in plant–environment interactions to adapt to unfavorable external environments [58]. How plants adapt to external abiotic stresses involves a very complex gene regulatory network, and a variety of transcription factors are usually involved in this process [58,59,60]. In previous studies, we found that *WRKY* and *MYB* genes were involved in regulating the response of hybrid *Liriodendron* to stress [56,57]. Therefore, we constructed the co-expression network based on the *LcAAAP*, *LcMYB*, and *LcWRKY* genes to explore the interactions between them. Multiple MYB and WRKY genes showed a significantly positive correlation, and *LcWRKY5/27* and *LcMYB9/28/83* could rise to a relatively high level in a short time under abiotic stress [56,57]. These results indicated that the *LcLHT1*, *LcAPP4a*, and *LcATL5a* could transport amino acids under abiotic stress, provide support for the internal homeostasis of plants, and then adapt to changes in the external environment, and the specific expression pattern of these genes was directly or indirectly regulated by upstream transcription factors.

Gamma-aminobutyric acid (GABA) enhances aluminum tolerance in hybrid *Liriodendron* [61]. Therefore, we explored the gene expression pattern of *LcGAT* and *LcPorT*, where *GAT* can specifically transport GABA and the PorT can barely transport GABA in addition to its ability to transport proline [34,36]. The *LcGAT1a/1c/2/4* were significantly upregulated in different organs after Al stress treatment; and the treatment of exogenous GABA causes GATs to respond at different rates in different organs. For example, the expression level of *LcGAT1c/2* in roots was gradually increased from 12 to 48 h, whereas it was gradually downregulated in buds. When exogenous GABA treatment was applied to Al stressed plants, the expression levels of *LcGAT1/2* in stems and leaves were rapidly upregulated within 12 h, indicating that this treatment may accelerate the transport of GABA in plants, thereby revealing that the treatment might accelerate the transport of GABA in the plant, more quickly alleviating the effect of Al poisoning on the growth and development of hybrid *Liriodendron*. For example, the accumulation of GABA in leaves facilitates the closure of plant stomata, thereby increasing its tolerance [62]. In addition, proline accumulated in hybrid *Liriodendron* leaves under the effect of Al toxicity based on previous studies; therefore, the expression level of *LcPorTs* showed different degrees of upregulation under aluminum stress treatment. *LcPorT1/2* showed opposite expression patterns in stems, leaves, and buds under GABA treatment; this indicates that there is also a relationship between the transport of *LcPorTs* and GABA, but the specific mode and pattern need further research.

## 4. Materials and Methods

### 4.1. Collection of Query Sequences and Identification of the AAAP Gene Family Based on L. chinense Genomic Data

At*AAAP* gene family protein sequences were acquired from the UniProT database (https://www.uniprot.org/ (accessed on 12 February 2022)) [63], and all the *AAAP* gene family sequences of other species were obtained from the JGI (https://phytozome.jgi.doe.gov/pz/portal.html (accessed on 13 February 2022)) [64]. The index sequences consisted of 46 At*AAAP* protein sequences based on the BLASTP. The 69 putative sequences were found from the *L. chinense* protein database through a local BLASTP search with an E-value threshold of e^−10^. The Hidden Markov mode (HMM) profile for the *AAAP* domain (PF01490) downloaded from the Pfam database (http://pfam.xfam.org (accessed on 15 February 2022)) [65], was used to identify *LcAAAP* genes from the *L. chinense* genome with HMMER 3.3.2 (http://hmmer.janelia.org/ (accessed on 15 February 2022)), with an E-value of e^−5^. A total of 73 putative sequences were identified from the genomic database. After all putative *LcAAAP* protein sequences were merged, the candidate sequences were further analyzed with the online tools Conserved Domain Database (http://www.ncbi.nlm.nih.gov/cdd/ (accessed on 16 February 2022)) [66] and pfam [67]. Finally, the 52 target sequences were identified and used for further analysis. Molecular weight and isoelectric point of each protein were determined using the online tool ExPASy (https://web.expasy.org/protparam/ (accessed on 17 January 2022)) [68]. The subcellular localization of all *LcAAAP* proteins was predicted based on the online tool Cell-PLoc 2.0 (http://www.csbio.sjtu.edu.cn/bioinf/Cell-PLoc-2/ (accessed on 17 January 2022)) [69].

### 4.2. Phylogenetic Tree, Gene Structure, and Conserved Motif and Transmembrane Region Analyses of LcAAAP Genes

Multiple sequence alignments analyses of *AAAP* amino acid sequences of *Arabidopsis thaliana*, *Oryza sativa*, *Vitis vinifera*, *Sorghum bicolor*, *Populus trichocarpa*, *Zea mays*, *Amborella trichopoda*, and *L. chinense* were performed with Muscle 3.8.31 [70]. We built the phylogenetic tree using the Maximum-Likelihood estimation with IQtree2.13 [71] and 1000 bootstrap replications; the optimal model was JTT+F+R7. The method for constructing the phylogenetic tree from the LcAAP protein sequences was consistent with the above; the optimal model was VT+F+G4. Evolutionary tree beautification was carried out using online tools iTOL (https://itol.embl.de/tree/2182103287321642042781 (accessed on 16 April 2022)) [72] and Adobe Illustrator CS3 software (version 13.0.0).

The exon/intron organization of *LcAAAP* genes was conducted by scanning the genomic annotation data. Conserved motifs were predicted with MEME (http://meme-suite.org/tools/meme (accessed on 20 February 2022)) [73] and a maximum of 20 motifs. The gene structure and motifs were visualized with TBtools [74]. Protein transmembrane topology was predicted using the ΔG prediction server v1.0 (https://dgpred.cbr.su.se/index.php?p=home (accessed on 16 April 2022)) [75]. Images were produced using the Tbtools software [74].

### 4.3. Chromosomal Location, Syntenic Analyses and Calculation of the Ka/Ks Value

The positions of the *LcAAAP* genes were acquired from the annotation file. Among them, 48 *LcAAAP* genes were mapped on 14 chromosomes, and the remaining four genes were located on the three contigs, with results listed in Table 1. We used MCScanX [76] with the default settings to identify *LcAAAP* gene pairs of segmental/tandem duplications in the *L. chinense* genome and analyzed the *LcAAAP* genes’ syntenic relation with the *AtAAAP*, *OsAAAP*, and *VvAAAP* gene pairs. The KaKs_Calculator2.0 [77] software was used to calculate the Ka/Ks values of homologous genes.

### 4.4. Analysis of the Codon Usage Pattern and The Cis-Acting Element

We obtained the coding sequences of *L. chinense*, *Arabidopsis thaliana*, *Oryza sativa*, and *Amborella trichopoda* based on the above *AAAP* genes. The CodonW1.4.4 software [78] was used to calculate the relative synonymous codon usage (RSCU) and the relative frequency of synonymous codons (RFSC). As promoter sequences, 2500 bp upstream from the translation start sites for *LcAAAP* genes were considered, and the cis-acting elements were predicted and analyzed using PlantCARE (http://bioinformatics.psb.ugent.be/webtools/plantcare/html/ (accessed on 16 April 2022)) [79].

### 4.5. RNA-Seq Analysis of LcAAAP Gene Expression Levels in Different Organs and Multiple Stresses

To explore the expression patterns of the *LcAAAP* gene family, transcript data of different organs and hybrid *Liriodendron* at high temperature (0 h, 1 h, 3 h, 6 h, 12 h) and drought (0 h, 1 h, 3 h, 6 h, 12 h, 24 h, and 72 h) stress data were downloaded from NCBI with the following accession numbers: SRR8101040, SRR8101041, SRR8101042, SRR8101043, SRR9945429, SRR9945430, SRR9945433, SRR9948913, SRR9948914, SRR9948915, SRR9948916, SRR9948917, SRR9948918, SRR9948919, SRR9949005, SRR9949006, SRR9949007, SRR9949008, SRR9949009, SRR9949010, PRJNA679089, and PRJNA679101. The low temperature (4 °C (0 h, 12 h, 24 h, and 48 h)), *Liriodendron* petal development (Lc1-4 and Lt1-4 represent the developmental process of the middle part of the petals of *L. chinense* and *L. tulipifera*, respectively) and Hybrid *Liriodendron* somatic embryogenesis (PEM: embryogenic callus; ES1: 10 days after liquid culture; ES2: 2 days after screening; ES3: ABA 1 day of treatment; ES4: ABA treatment for 3 days; ES5: globular embryo; ES6: heart-shaped embryo; ES7: torpedo embryo; ES8: immature cotyledon embryo; ES9: mature cotyledon embryo; PL: plantlet) transcript data own the undisclosed data; The expression levels of related genes were listed in Appendix A. All mRNA abundance values were measured by transcripts per million (TPM) based on the *L. chinense* genomic database.

### 4.6. Plant Materials and Stress Treatment

According to the previous research, the *GAT* and *ProT* could transport the gamma-aminobutyric acid (GABA), and GABA signaling could enhance the tolerance of hybrid *Liriodendron* to Al stress by promoting organic acid transport and maintaining cellular redox and osmotic balance [61]. To compare the *GAT* and *ProT* subgroup expression levels in organs of different stages in the Al stress, we chose the one-month hybrid *Liriodendron* with basically the same growth for stress treatment (Nanjing, China). The cultivation environment is the greenhouse of the Key Laboratory of Forest Genetics and Biotechnology of the Ministry of Education, Nanjing Forestry University, under white light (light for 16 h, darkness for 8 h). The concentrations of AlCl_3_ and GABA were 30 uM and 10 mM, respectively; the culture medium was 3/4 MS. Hybrid *Liriodendron* seedlings were soaked in Al solution for 3h and then inoculated on the corresponding medium; we obtained materials from three time periods, 12 h, 24 h, and 48 h, respectively. We quickly froze the material in liquid nitrogen and stored it at −80 °C.

### 4.7. RNA Extraction and Quantitative Real-Time PCR Analysis

Total RNAs were extracted using the KK Fast Plant Total RNA Kit (ZOMANBIO, Beijing, China). The Evo M-MLV RT Kit with gDNA Clean for qPCRII AG11711 (Accurate Biotechnology (Hunan) Co., Ltd., Changsha, China) was applied to synthesize the first-strand cDNA from 1.0 mg RNA. Equalbit 1× dsDNA HS Assay Kit (EQ121-01, vazyme) completed quantification of all reversed cDNAs. PCR amplifications were carried out with SYBR^®^ Green Premix Pro Taq HS qPCR Kit (Accurate Biotechnology (Hunan) Co., Ltd., Changsha, China) in 20 μL volumes using Roche LightCycler^®^ 480 Real-Time PCR System. Three replicates were performed for each selected gene. The 18S was taken as a reference gene. The relative expression levels were calculated by the ∆∆ CT method [80]. All qRT-PCR primers were designed by Primer5.0 [81] and are listed in Appendix A.

## 5. Conclusions

In this study, we identified 52 *AAAP* genes from the *L. chinense* genome. Multiple analyses, including the construct phylogenetic tree, gene exon–intron structure, motif, and transmembrane domain prediction, indicated that the *LcAAAP* gene family was divided into eight subgroups. Most of the identified *LcAAAP* proteins have a close evolutionary relationship with *A. trichopoda* proteins, implying that they were relatively primitive among angiosperms. WGD and gene duplication events suggested that the expansion of some subgroups during the evolutionary history led to divergent gene functions. The expression patterns in different organs showed that the *LcAAAPs* were specifically expressed in the nectary development and somatic embryogenesis of *Liriodendron*, indicating that the *LcAAAP* gene family has coordinated to regulate growth and development. In addition, the *LcAAAP* gene family can also respond to low temperature, heat, and drought stresses through the regulation of *WRKY/MYB* genes; and the members of the *GAT* and *PorT* subgroups members can participate in the process of hybrid *Liriodendron* to resist Al poisoning.

## Figures and Tables

**Figure 1 ijms-23-04765-f001:**
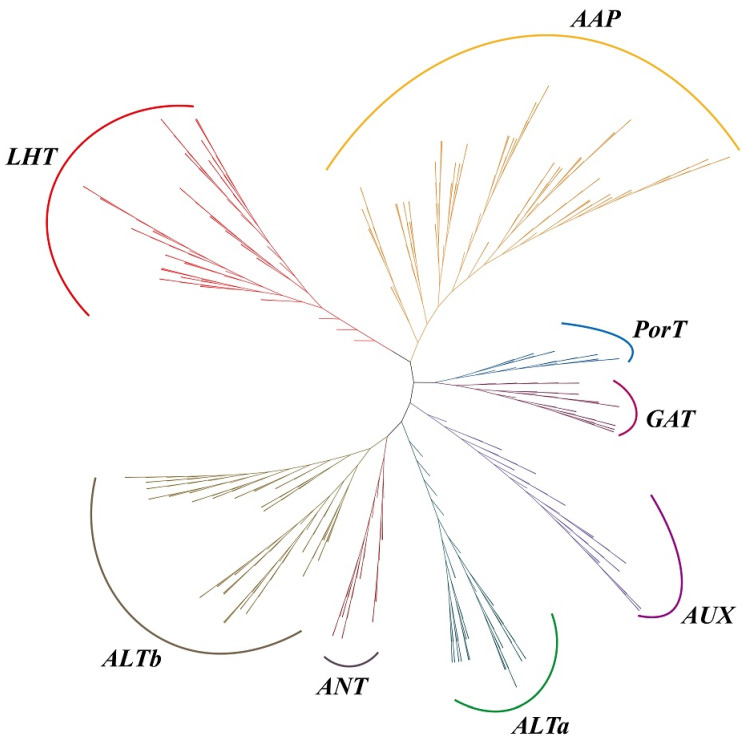
Phylogenetic tree of *L. chinense* (Lc), *O. sativa* (Os), *A. thaliana* (At), *A. trichopoda* (Atr), *S. bicolor* (Sb), *P. trichocarpa* (Pt), *Z. mays* (Zm), and *V. vinifera* (Vv) *AAAP* proteins. Multiple sequence alignment of full-length proteins was performed by muscle and the phylogenetic tree using the maximum-likelihood estimation with IQtree2.13 and 1000 bootstrap replications: the optimal model was JTT+F+R7. The tree was divided into eight subgroups, marked by different-colored backgrounds.

**Figure 2 ijms-23-04765-f002:**
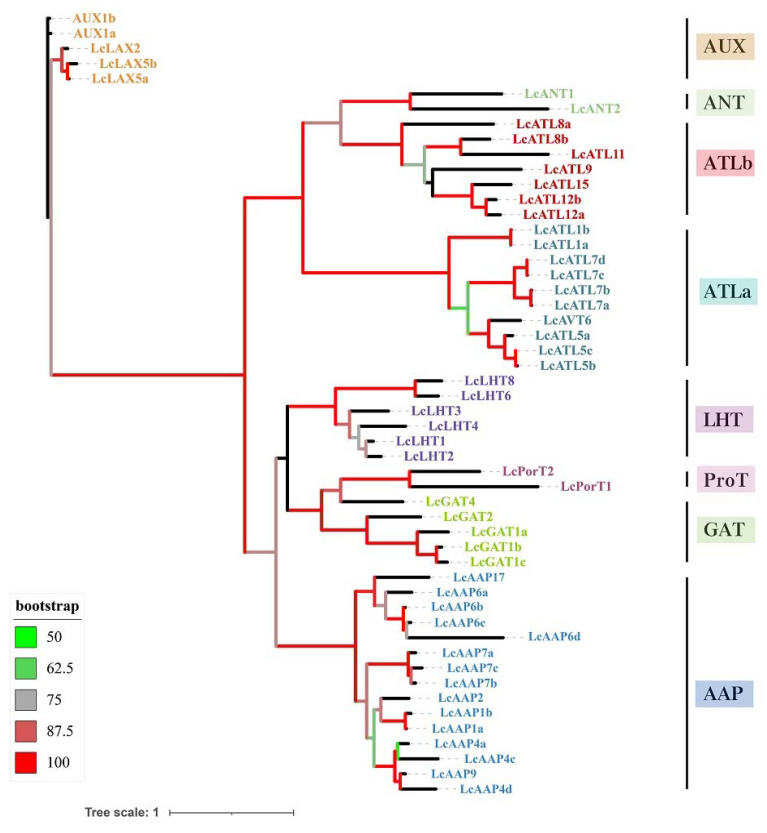
Phylogenetic relationships of the *AAAP* gene family in *L. chinense*. Multiple sequence alignment of full-length proteins was performed by muscle and the phylogenetic tree using the maximum-likelihood estimation with IQtree2.13 and 1000 bootstrap replications: the optimal model was VT+F+G4. The tree was divided into 8 subgroups, marked by different-colored backgrounds.

**Figure 3 ijms-23-04765-f003:**
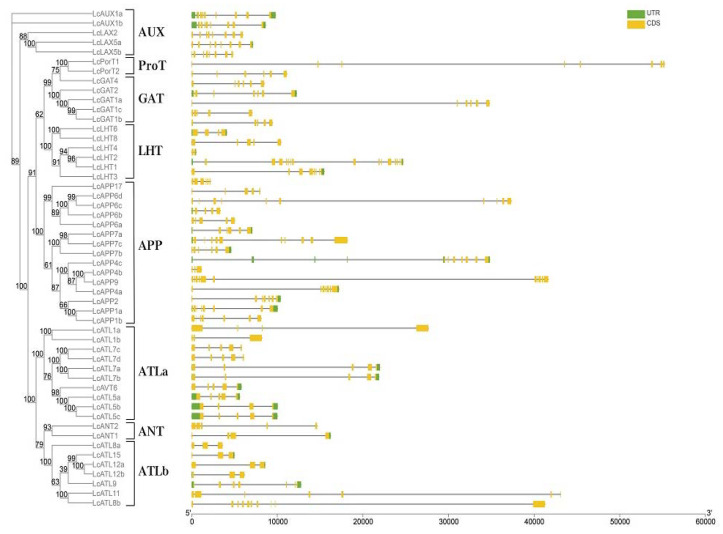
Phylogenetic relationship and gene structures of *LcAAAP*s. Phylogenetic tree of 52 *L**cAAAP*s proteins. Maximum-Likelihood tree was constructed using IQtree. Bootstrap support values from 1000 reiterations are indicated at each node. The 52 *LcAAAP*s in the tree were divided into eight subfamilies. Gene structure was indicated by green and yellow rectangles, respectively.

**Figure 4 ijms-23-04765-f004:**
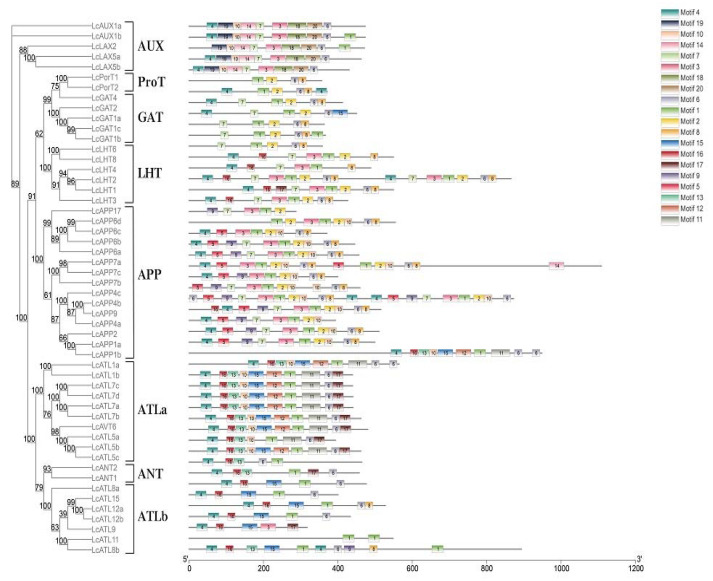
Phylogenetic relationship and conserved motifs of *LcAAAP*s. Phylogenetic tree of 52 *L**cAAAP*s proteins. Conserved motifs of *LcAAAP*s proteins. Each colored box represents a specific motif in the protein identified using the MEME motif search tool. The order of the motifs corresponds to their position within individual protein sequences.

**Figure 5 ijms-23-04765-f005:**
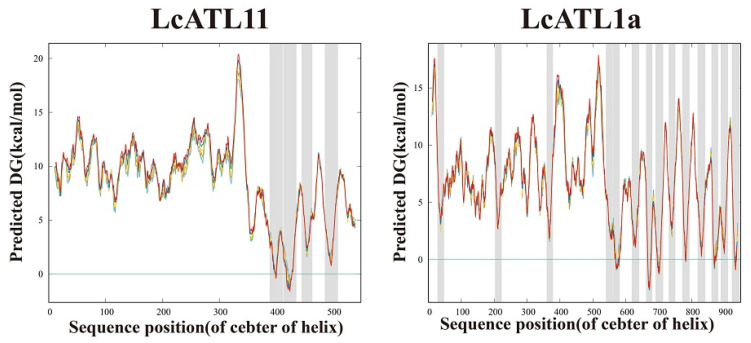
Prediction results of the transmembrane domain of *LcATL1a*, *LcLHT2*, *LcAPP4b*, *LcATL11*, *LcAAP17*, *LcATL9*, and *LcPorT1* proteins. The grey area represents the putative transmembrane helix.

**Figure 6 ijms-23-04765-f006:**
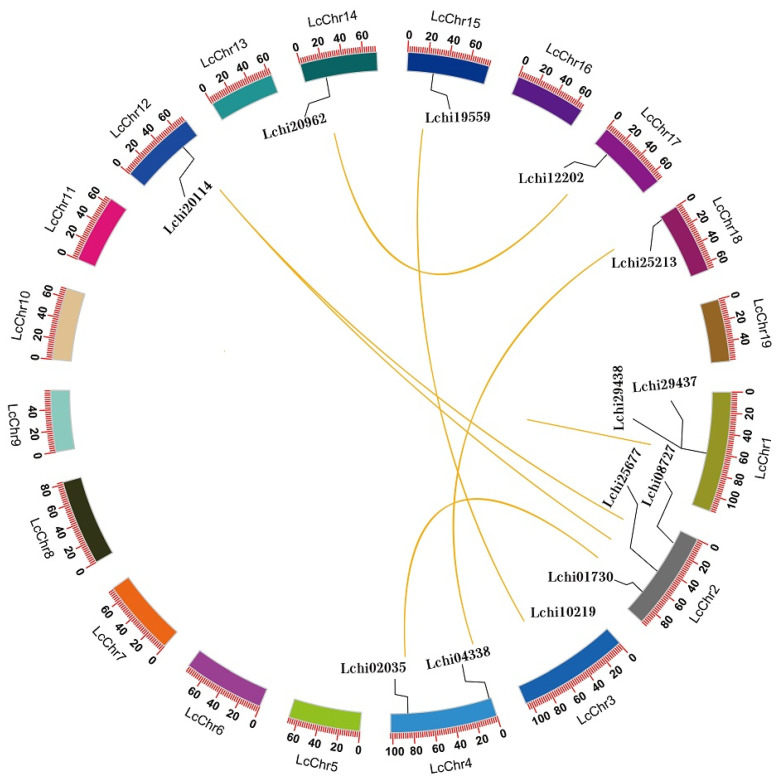
Distribution and segmental duplication of *LcAAAP* genes in *L. chinense*. The different colors in the panel show the chromosomes using a circle. Yellow lines connect homologous genes; chromosome numbers are marked outside of the circle.

**Figure 7 ijms-23-04765-f007:**
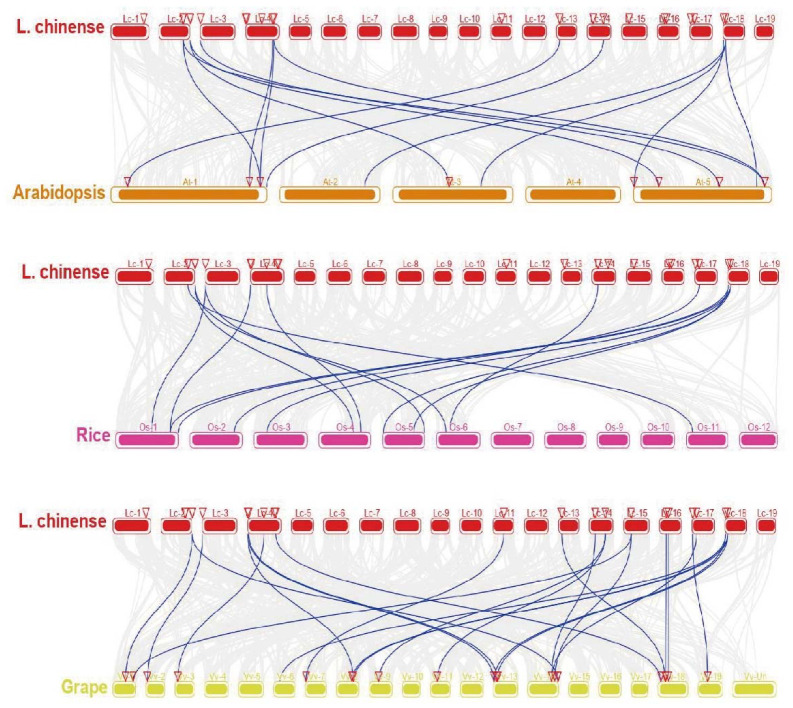
Genome-wide synteny analysis of *LcAAAP* gene family among *L. chinense* and three other species. Synteny analysis of *LcAAAP* genes between *L. chinense*, Arabidopsis, rice, and grape. Gray lines in the background indicate the collinear blocks between *L. chinense*, Arabidopsis, rice, and grape genomes, while the blue lines highlight the syntenic *AAAP* gene pairs. Red open triangles represent homologous genes.

**Figure 8 ijms-23-04765-f008:**
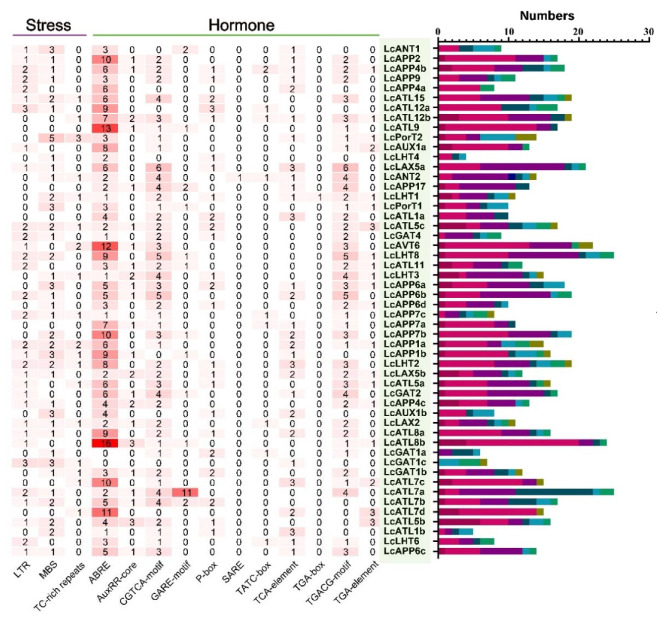
The number of cis-elements in *LcAAAP* promoters. The motif types of hormones and stress responsiveness are shown in green and purple lines in *LcAAAP* gene families. Red represents the quantity. The bar chart on the right represents the number of response elements contained in each gene, and different colors represent different types of elements.

**Figure 9 ijms-23-04765-f009:**
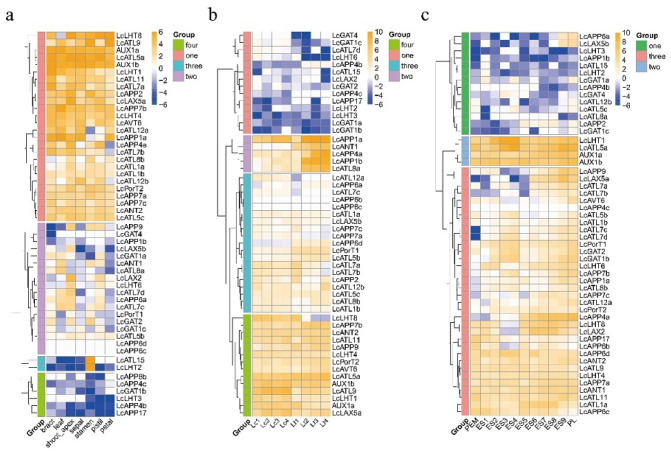
The *LcAAAP* genes expression profiles in different organs. The heatmap shows the mean of three biological replicates. Transcripts per million (TPM) was used to indicate the gene expression level. (**a**–**c**): The expression level of *LcAAAP* genes in different organs, nectary development, and somatic embryogenesis of hybrid *Liriodendron*; Lc1-4 and Lt1-4 represent the developmental process of the middle part of the petals of *L. chinense* and *L. tulipifera*, respectively. PEM: embryogenic callus; ES1: 10 days after liquid culture; ES2: 2 days after screening; ES3: ABA 1 day of treatment; ES4: ABA treatment for three days; ES5: globular embryo; ES6: heart-shaped embryo; ES7: torpedo embryo; ES8: immature cotyledon embryo; ES9: mature cotyledon embryo; PL: plantlet.

**Figure 10 ijms-23-04765-f010:**
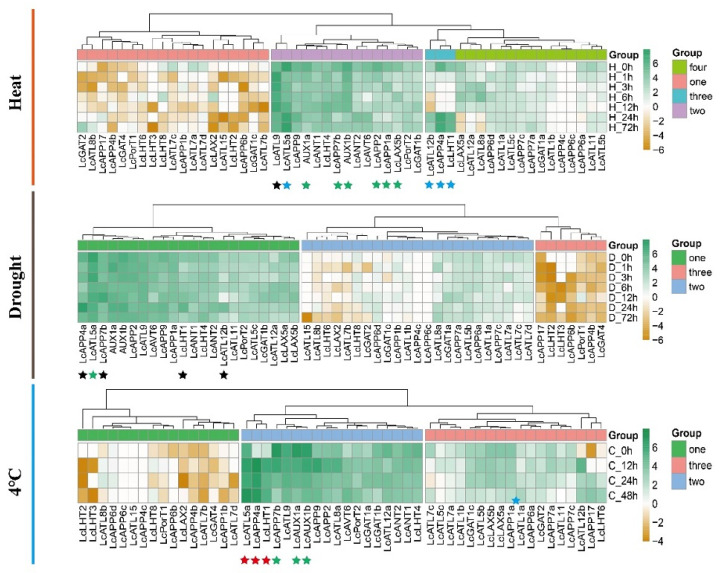
The *LcAAAP* genes expression profiles in different stress. The heatmap shows the mean of three biological replicates. Transcripts per million (TPM) was used to indicate the gene expression level. Expression patterns of *LcAAAP* genes in leaves of hybrid *Liriodendron* under heat (40 °C), low temperature (4 °C), and drought stress. Stars of different colors represent different gene expression patterns, black represents up-regulation and then down-regulation, green represents down-regulation, blue represents down-regulation and then up-regulation, and red represents up-regulation.

**Figure 11 ijms-23-04765-f011:**
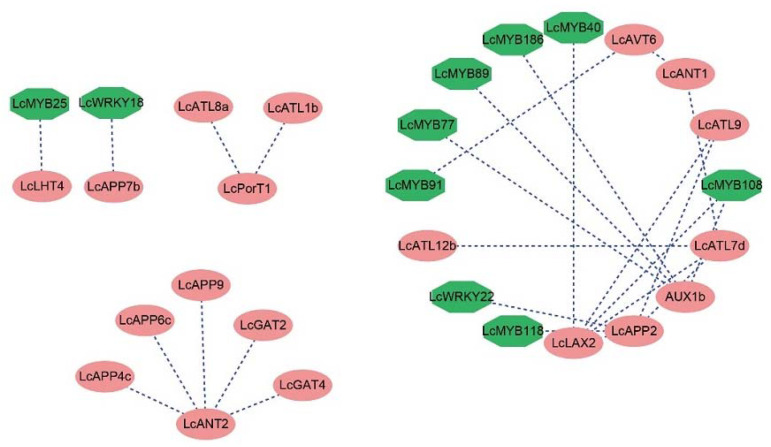
Prediction network of protein interactions for *LcAAAP*, LcWRKY, and LcMYB gene families. Pink ovals represent *AAAP* gene family members, dark green hexagons represent MYB/WRKY transcription factors, and dark blue dashed lines represent predicted correlations.

**Figure 12 ijms-23-04765-f012:**
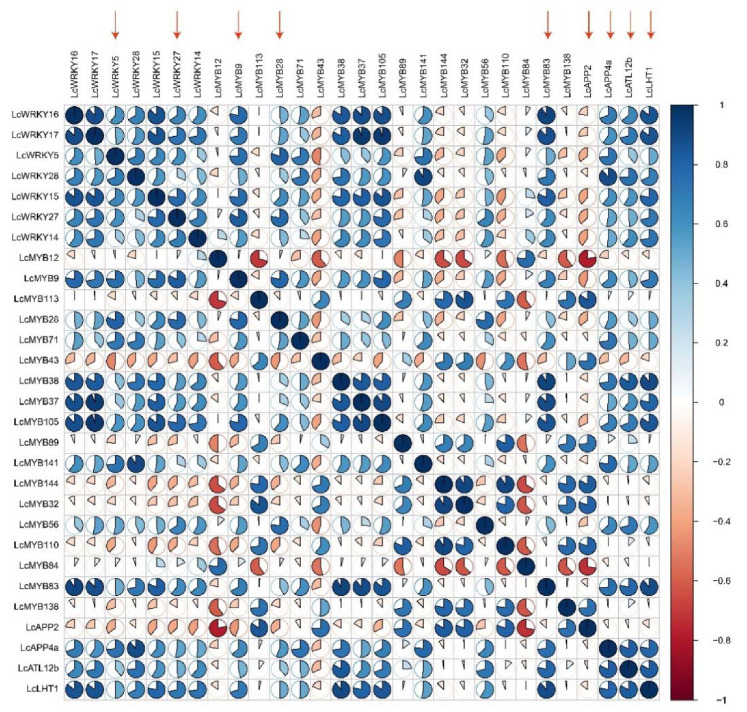
Correlation analysis of *LcAAAP*/*LcWRKY*/*LcMYB* genes under drought, low temperature, and heat stress. The red arrows indicate that these genes were significantly upregulated under the three stressors.

**Figure 13 ijms-23-04765-f013:**
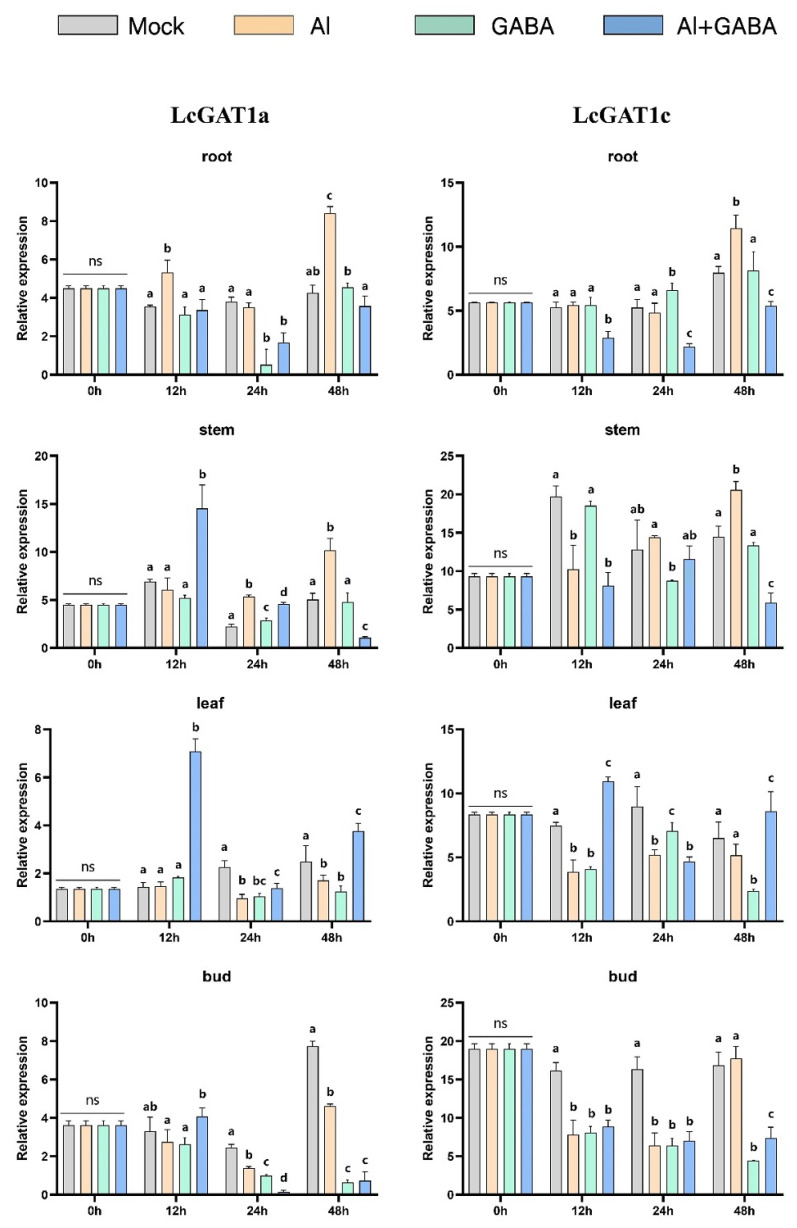
Gene expression of *LcGAT*1 in hybrid *Liriodendron* under Al toxicity and GABA treatment. Mock represents the ultrapure water treatment; Al represents the aluminum stress treatment; GABA represents exogenously applied GABA treatment; Al+GABA represents Al stress treatment and then exogenously applied GABA treatment.

**Figure 14 ijms-23-04765-f014:**
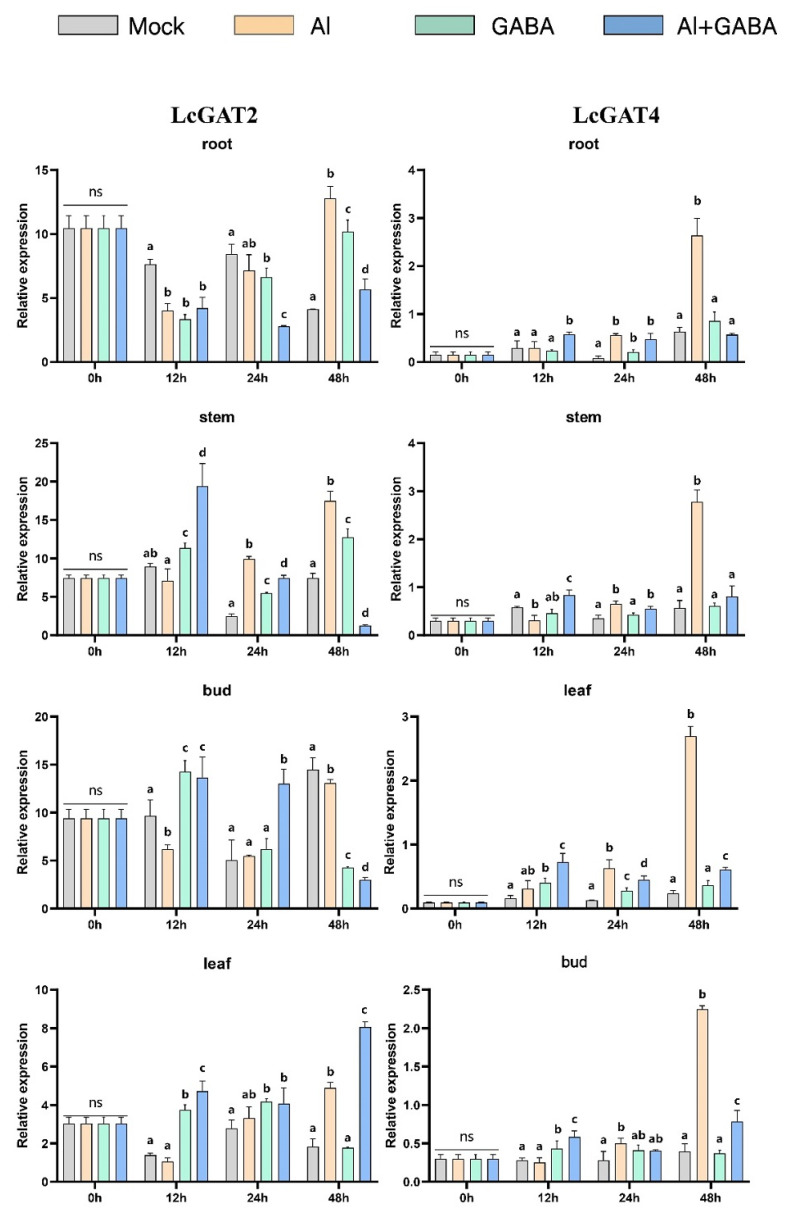
Gene expression of *LcGAT**2* and 4 in hybrid *Liriodendron* under Al toxicity and GABA treatment. Mock represents the ultrapure water treatment; Al represents the aluminum stress treatment; GABA represents exogenously applied GABA treatment; Al+GABA represents Al stress treatment and then exogenously applied GABA treatment.

**Figure 15 ijms-23-04765-f015:**
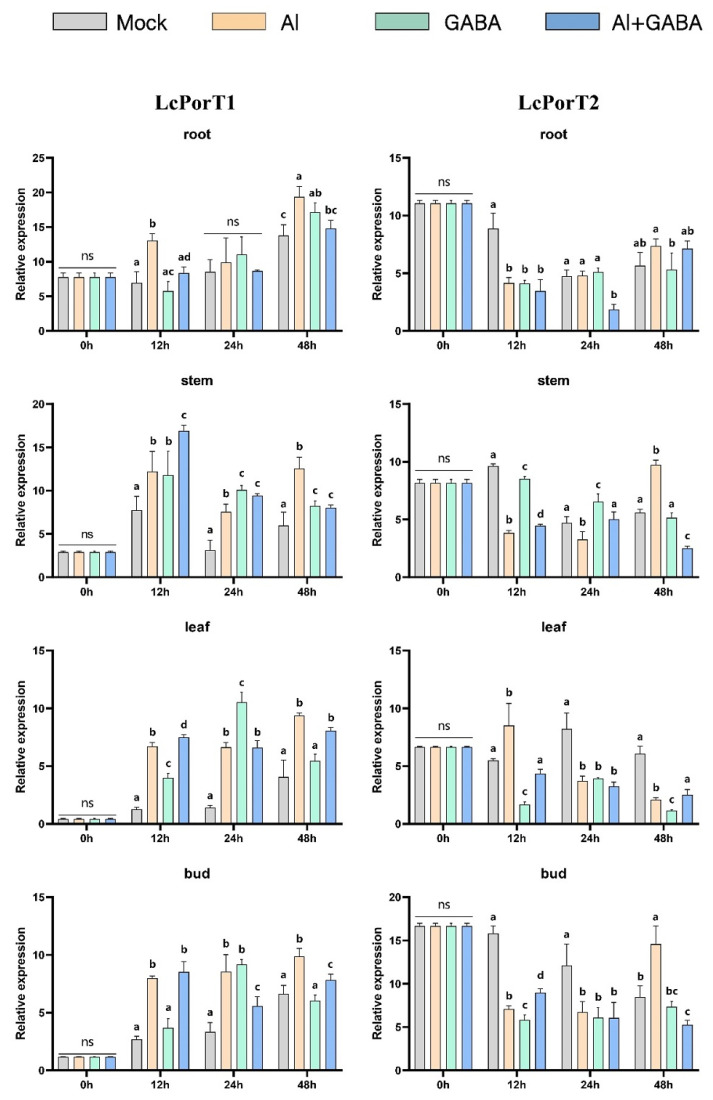
Gene expression of *LcPorT* in hybrid *Liriodendron* under Al toxicity and GABA treatment. Mock represents the ultrapure water treatment; Al represents the aluminum stress treatment; GABA represents exogenously applied GABA treatment; Al+GABA represents Al stress treatment and then exogenously applied GABA treatment.

**Table 1 ijms-23-04765-t001:** The information of 52 *LcAAAP* genes in *L. chinense*.

Gene ID	Gene Name	Locus	Location	ORF (bp)	Amino Acid Length	MW (KDa)	pI	Subcellular Localization
*Lchi00592*	*LcANT1*	*chr1*	108060548	1377	458	49.91	7.98	Cell membrane. Chloroplast.
*Lchi00907*	*LcAPP2*	*chr1*	98782372	1185	394	43.23	9.15	Cell membrane. Golgi apparatus.
*Lchi29437*	*LcGAT1a*	*chr1*	64003591	1095	364	39.33	9.65	Cell membrane
*Lchi29438*	*LcGAT1c*	*chr1*	63956981	1104	367	39.96	9.54	Cell membrane
*Lchi29446*	*LcGAT1b*	*chr1*	63673692	1080	359	38.92	9.7	Cell membrane
*Lchi01730*	*LcAPP4b*	*chr2*	72696256	2622	873	97.32	9.5	Cell membrane
*Lchi01732*	*LcAPP9*	*chr2*	72772418	1551	516	56.63	9.23	Cell membrane
*Lchi02902*	*LcATL15*	*chr2*	93230749	1206	401	43.99	9.03	Cell membrane. Golgi apparatus.
*Lchi02903*	*LcATL12a*	*chr2*	93204648	1587	528	58.45	9	Cell membrane
*Lchi02904*	*LcATL12b*	*chr2*	93165315	1305	434	47.18	7.98	Cell membrane. Golgi apparatus.
*Lchi08727*	*LcLAX5a*	*chr2*	16848262	1392	463	52.34	8.95	Cell membrane
*Lchi25677*	*LcLAX2*	*chr2*	45173338	1419	472	53.49	9.09	Cell membrane
*Lchi10219*	*LcLHT1*	*chr3*	67706260	1653	550	60.34	8.74	Cell membrane
*Lchi10341*	*LcPorT1*	*chr3*	75189713	1074	357	40.21	9.9	Cell membrane
*Lchi22116*	*LcGAT2*	*chr3*	904020	1356	451	49.53	8.87	Cell membrane. Golgi apparatus.
*Lchi02035*	*LcAPP4a*	*chr4*	84533298	1452	483	53.24	9.02	Cell membrane
*Lchi04257*	*LcPorT2*	*chr4*	3186704	1119	372	40.72	9.36	Cell membrane
*Lchi04338*	*LcAUX1a*	*chr4*	642668	1425	474	53.43	8.68	Cell membrane
*Lchi09378*	*LcANT2*	*chr4*	48992807	1398	465	50.56	6.41	Cell membrane
*Lchi10104*	*LcAPP17*	*chr4*	78821859	867	288	31.32	6.3	Cell membrane
*Lchi16175*	*LcLHT3*	*chr4*	80850467	1284	427	47.47	8.73	Cell membrane. Chloroplast. Golgi apparatus.
*Lchi16222*	*LcAPP6a*	*chr4*	79369615	1374	457	50.52	9.01	Golgi apparatus.
*Lchi16226*	*LcAPP6b*	*chr4*	79186487	1341	446	49.25	9.39	Golgi apparatus.
*Lchi16229*	*LcAPP6d*	*chr4*	79108510	1668	555	62.66	6.44	Cell membrane. Chloroplast.
*Lchi29396*	*LcATL8b*	*chr4*	11575526	2685	894	97.75	5.56	Cell membrane
*Lchi33844*	*LcLHT6*	*chr4*	28079397	1653	550	60.5	9.49	Cell membrane
*Lchi05928*	*LcLHT4*	*chr8*	74276594	1209	402	44.78	8.94	Cell membrane. Golgi apparatus.
*Lchi14454*	*LcATL11*	*chr10*	29873111	1650	549	61.75	7.53	Cell membrane. Cytoplasm. Mitochondrion. Nucleus.
*Lchi13365*	*LcGAT4*	*chr11*	32708594	1242	413	45.04	9.34	Cell membrane
*Lchi23105*	*LcAPP4c*	*chr11*	28298237	1239	412	45.75	8.7	Cell membrane
*Lchi20114*	*LcLAX5b*	*chr12*	55555547	1296	431	48.96	9.12	Cell membrane
*Lchi18276*	*LcAPP1a*	*chr13*	8635305	1536	511	56.19	8.83	Cell membrane
*Lchi18277*	*LcAPP1b*	*chr13*	8656388	1503	500	55.15	8.91	Cell membrane
*Lchi03159*	*LcATL9*	*chr14*	62637871	957	318	34.88	9.19	Cell membrane
*Lchi10700*	*LcATL1a*	*chr14*	48691940	2850	949	103.81	6.41	Cell membrane
*Lchi20962*	*LcATL5a*	*chr14*	17705266	1446	481	51.79	8.84	Cell membrane. Chloroplast
*Lchi32419*	*LcATL1b*	*chr14*	36109607	1698	565	61.58	8.49	Chloroplast.
*Lchi19559*	*LcLHT2*	*chr15*	22627745	2601	866	96.38	8.97	Cell membrane
*Lchi13826*	*LcAVT6*	*chr16*	20708277	1392	463	49.39	7.01	Cell membrane. Chloroplast. Cytoplasm. Golgi apparatus.
*Lchi13928*	*LcLHT8*	*chr16*	27398496	1470	489	54.17	9.56	Cell membrane
*Lchi12202*	*LcATL5c*	*chr17*	13443833	1389	462	49.68	7.62	Cell membrane. Golgi apparatus.
*Lchi16887*	*LcAPP7c*	*chr17*	2623000	1203	400	44.08	6.41	Cell membrane
*Lchi16888*	*LcAPP7a*	*chr17*	2654807	3330	1109	123.62	7.78	Cell membrane. Cytoplasm.
*Lchi16889*	*LcAPP7b*	*chr17*	2708322	1383	460	51.9	9.16	Cell membrane
*Lchi32115*	*LcATL5b*	*chr17*	53273477	1185	394	42.31	5.94	Cell membrane
*Lchi25213*	*LcAUX1b*	*chr18*	2575200	1425	474	53.49	8.68	Cell membrane
*Lchi30378*	*LcATL7c*	*chr18*	56630104	1323	440	47.12	6.14	Cell membrane. Chloroplast. Cytoplasm. Golgi apparatus.
*Lchi30379*	*LcATL7a*	*chr18*	56674114	1326	441	47.66	8.24	Cell membrane. Chloroplast. Golgi apparatus.
*Lchi34889*	*LcAPP6c*	*Contig1773*	10550	1116	371	41.41	9.85	Cell membrane
*Lchi31213*	*LcATL7b*	*Contig2824*	600473	1326	441	47.64	8.56	Cell membrane. Chloroplast. Golgi apparatus.
*Lchi31214*	*LcATL7d*	*Contig2824*	620344	1323	440	47.1	6.14	Cell membrane. Chloroplast. Cytoplasm. Golgi apparatus.
*Lchi28875*	*LcATL8a*	*Contig509*	1166972	1434	477	51.37	9.13	Golgi apparatus.

## Data Availability

The datasets supporting the conclusions and description of a complete protocol can be found within the manuscript and its additional files. The datasets used and/or analyzed during the current study are available from the corresponding author on reasonable request.

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
