# Peer review of "Identification, Phylogenetic and Expression Analyses of the *AAAP* Gene Family in *Liriodendron chinense* Reveal Their Putative Functions in Response to Organ and Multiple Abiotic Stresses"

_ijms, 2022, doi:10.3390/ijms23094765_

Round 1

Reviewer 1 Report

Dear Authors,

I had a great opportunity to review manuscript entitled: “Identification, phylogenetic and expression analyses of the AAAP gene family in L. chinense reveal their putative functions  in response to tissue and multiple abiotic stresses” which is considered for publication  in IJMS journal.

The article presents interesting new insights in role of AAAP gen family, however, it has some problems (like problems with scientific literature), which need to be improved. List of specific comments are presented below:

1.Introduction section:

According IJMS publication rules this part should first introduce the problem and ended with precisely formulated aim of the study and eventually hypothesis. Currently introduction has no information about  L. chinense , why this plant is important in context of scientific research?. Currently the meaning/importance of research is unknown. Authors presents a lot of data in other plants (like Arabidopsis) but no data about the characterizations AAAP gene family in plants more phylogenetic connected with L. chinense. Moreover, Authors did not use full name of plant or the name of botanic family, in any point, in which this plant is classified. The other problem is that this section did not has any precisely formulated hypothesis and aim of the study. Mentioned above elements should be updated in introduction section.

  1. Results section

This part is the most problematic part. First of all, Authors must rethink, which results is truly important and/or  which could be omitted or transferred into supplementary data. Reason of that is that amount of data and the quantity of data on figures made them completely chaotic and unreadable. This lower the quality of manuscript and the quality of research. Some figures are completely too small or bad prepared and completely unreadable. Problem is contacted with:

Figure 1. Presents too many plants in the contest of phylogenetic analyses. The results is completely unreadable (too small or low quality of phylogenetic tree) and because of that fact the results is useless. The phylogenetic analyzes performed once more with less number of plant species (carefully and logical selected).

I strongly recommend to use for phylogenetic analyses plants, which are in a growth form and phylogenetically more closed to  L. chinense . Is little illogical to use of Arabidopsis, Zea or Oryza, because these plants are completely different in growth form and genome itself.

Table 1. The description of this table should be above table not in it. Moreover, the description of the table should have information about the source of information for example in which program or method authors performed the subcellular localization prediction. The data in the table should be also logically segregated, for example, by the chromosome localization all genes on ch1 then all gens on chr2 etc. Or segregated by the type of genes like AUX, then ANT etc. Now the table and information is little chaotic.

Figure 3. should be separated for two separate Figures. After that they should be enlarged as big as possible now is not readable and in low quality;

Figure 4. The figure descriptions is crossed with main figure, please correct it. The Figure should be separated to 3 figures, now the results is too low quality and unreadable. After separation they should be enlarged as big as possible. Authors need to add in the description information about program used for prediction in my opinion it is TMHMM 2.0? but maybe other method was selected? I suggested also add analyses for selected proteins associated with membranes elements like TGN (Trans Golgi network) in ΔG prediction server v1.0 . ΔG is standard tool for confirmation or excluding results for standard transmembrane prediction.

Figure 5 to small/low quality and unreadable

Table 2 and 3 should be moved into supplementary data

Figure 6 to small/low quality and unreadable

Figure 7. to small/low quality and unreadable. Bad pixels nothing could be observed. Authors claimed that a presents different tissues expression which tissues they have in mind? Plant tissues are epidermis, mesophyll, xylem and/or phloem, for example. Authors did not perform genes expression profiles for tissues at all. This is a huge scientific error and unfortunately also suggest that authors did not know what they exactly analysed.  Only based on barely legible  markings on figure indicate that authors  performed analyses of expression in different plant organs -not tissues.   The Authors must separate this figure to two separate figures (on for plant organ expression and one for abiotic stresses) and make them in HD quality.

Figure 8 The results presented in figure is completely unreadable;

Figure 10. The results presented in figure is completely unreadable. Authors should split the results. Charts and expression results or even statistical significance is unreadable.

  1. Materials and methods section

This part is also problematic. This section is too laconic. Materials and methods section should be presented in a form that enables the repetition of results. This section must be extended. This section has also small number of citations, or  some methods has not any citation but are methods performed widely on the world, but not developed by authors. So I do not understand the lack of citations? Moreover, all used programs with http addresses should be also cited as literature positions as IJMS publication rules suggest. The all used hardware, programs and solution need has information about producer and country of the producer.

  1. Reference section (minor)

Any of the positions from reference list is not prepared according IJMS strict publication rules. Moreover, references have many errors in names of journals or spelling. This part must be greatly improved.

Author Response

Dear editor and reviewers:

I am very grateful to your comments for the manuscript. According with your advice, we amended the relevant part in manuscript. Here below is our description on revision according to the reviewers’ comments.

Reviewer #1:

Point 1: Currently introduction has no information about L. chinense, why this plant is important in context of scientific research? Currently the meaning/importance of research is unknown.

Response 1: On the 29-31, we supplement the background content of related research on Liriodendron.

Point 2: Authors presents a lot of data in other plants (like Arabidopsis) but no data about the characterizations AAAP gene family in plants more phylogenetic connected with L. chinense. Moreover, Authors did not use full name of plant or the name of botanic family, in any point, in which this plant is classified.

Response 2: On the 65-96, due to the special evolutionary status of Liriodendron, which belongs to the basic angiosperm, there are few studies on related genes of its related species. The related research on the AAAP gene family related to other species such as tea, poplar, potato, cotton, ginseng, etc. is supplemented here, and the full Latin name of the related species is also supplemented.

Point 3: The other problem is that this section did not has any precisely formulated hypothesis and aim of the study.

Response 3: On the 98-106, we add research purpose and significance.

Point 4: Figure 1. Presents too many plants in the contest of phylogenetic analyses. The results are completely unreadable (too small or low quality of phylogenetic tree) and because of that fact the results is useless. The phylogenetic analyzes performed once more with less number of plant species (carefully and logical selected).

Response 4: Thank you very much for finding out this problem, which is a great help to the rigor of our article. As the basic angiosperm, the evolutionary status of L. chinense belong to the base of the monocotyledonous and dicotyledonous plants, therefore, we chose the aaap gene family of monocotyledonous (O. sativa (Os), S. bicolor (Sb), Z. mays (Zm)), dicotyledonous (A. thaliana (At), P. trichocarpa (Pt), V. vinifera (Vv)), and A. trichopoda (Atr) construct the phylogenetic tree. We provided the on the Fig1 and the specific information of subgroups were provided in the supplements Fig 1.

Point 5: Table 1. The description of this table should be above table not in it. Moreover, the description of the table should have information about the source of information for example in which program or method authors performed the subcellular localization prediction. The data in the table should be also logically segregated, for example, by the chromosome localization all genes on ch1 then all gens on chr2 etc. Or segregated by the type of genes like AUX, then ANT etc. Now the table and information are little chaotic.

Response 5: On lines 569-572, we provide online analysis website information on materials and methods, and the data were arranged by chromosomal location information.

Point 6: Figure 3. should be separated for two separate Figures. After that they should be enlarged as big as possible now is not readable and in low quality;

Response 6: The Fig 3 was divided into Fig3 and 4, and be enlarged.

Point 7: Figure 4. The figure descriptions were crossed with main figure, please correct it. The Figure should be separated to 3 figures, now the results is too low quality and unreadable. After separation they should be enlarged as big as possible. Authors need to add in the description information about program used for prediction in my opinion it is TMHMM 2.0? but maybe other method was selected? I suggested also add analyses for selected proteins associated with membranes elements like TGN (Trans Golgi network) in ΔG prediction server v1.0. ΔG is standard tool for confirmation or excluding results for standard transmembrane prediction.

Response 7: According to your suggestion, we used ΔG prediction server v1.0 to re-predict the transmembrane helix structure, among which we added the larger difference to the Fig5, while the transmembrane structure information of the whole family is placed in the appendix Fig S2 and Table S3 respectively

Point 8: Figure 5 to small/low quality and unreadable.

Response 8: We split Figure 5 into Figures 6, 7 and increased the resolution.

Point 9: Table 2 and 3 should be moved into supplementary data.

Response 9: Tables 2 and 3 are divided into Table S4 and S5.

Point 10: Figure 6 to small/low quality and unreadable

Response 10: We repainted Fig6 and increased the resolution as Fig8.

Point 11: Figure 7. to small/low quality and unreadable. Bad pixels nothing could be observed. Authors claimed that a presents different tissues expression which tissues they have in mind? Plant tissues are epidermis, mesophyll, xylem and/or phloem, for example. Authors did not perform genes expression profiles for tissues at all. This is a huge scientific error and unfortunately also suggest that authors did not know what they exactly analysed. Only based on barely legible markings on figure indicate that authors performed analyses of expression in different plant organs -not tissues.   The Authors must separate this figure to two separate figures (on for plant organ expression and one for abiotic stresses) and make them in HD quality.

Response 11: As required, we differentiated organ expression levels and stress-responsive expression levels as Fig9 and Fig10, and increased the resolution of the images. The relevant tissues expression content has been changed to organ expression patterns in the article.

Point 12: Figure 8 The results presented in figure is completely unreadable;

Response 12: We increased the resolution of Fig 8

Point 13: Figure 10. The results presented in figure is completely unreadable. Authors should split the results. Charts and expression results or even statistical significance is unreadable.

Response 13: We increased the resolution of Fig 10 as the Fig 13.

Point 14: The materials and methods section were also problematic. This section is too laconic. Materials and methods section should be presented in a form that enables the repetition of results. This section must be extended. This section has also small number of citations, or some methods has not any citation but are methods performed widely on the world, but not developed by authors. So, I do not understand the lack of citations? Moreover, all used programs with http addresses should be also cited as literature positions as IJMS publication rules suggest. The all used hardware, programs and solution need has information about producer and country of the producer.

Response 14: According to your request, we will further improve the content of materials and methods on the 548-647.

Point 15: Reference section (minor). Any of the positions from reference list is not prepared according IJMS strict publication rules. Moreover, references have many errors in names of journals or spelling. This part must be greatly improved.

Response 15: References updated according to journal format.

All the lines and pages indicated above are in the revised manuscript (Identification, phylogenetic and expression analyses of the AAAP gene family in Liriodendron chinense reveal their putative functions in response to organ and multiple abiotic stresses) and the revisions were marked with a blue background. Thank you again for asking this question, which is of great help to the arrangement of our experimental content and the writing logic of the article. We sincerely hope to receive your comments on our revised content again.

Sincerely yours,

Hu Lingfeng

Reviewer 2 Report

IJMS-1657553

In this manuscript, 52 AAAP genes were identified in the Liriodendron chinense genome and divided into eight subgroups according to several features, as phylogenetic relationships, gene structure, and conserved motif. Forty-eight LcAAAP genes were located on the 14 chromosomes, and the remaining 4 genes were mapped in the contigs. The LcAAAP gene family is closer to the AAAP of Amborella trichopoda, based on the multispecies phylogenetic tree and codon usage bias analysis, indicating that the LcAAAP gene family is relatively primitive in angiosperms. 

Interestingly, low temperature, drought, and heat stresses may activate some WRKY/MYB transcription factors to positively regulate the expression of LcAAAP genes to achieve long-distance transport of amino acids in plants to resist the unfavorable external environment. In addition, the GAT and PorT subgroups could involve gamma-aminobutyric acid (GABA) transport under aluminum poisoning. These characteristiics of resistance to environment and too toxic elements is of great importance for regulation and biotechnological applications.

The results obtained in this study, could represents a basis for further study of the biological role of gene LcAAAP and improvement of the stress resistance of Liriodendron

The manuscript is well written and represents an important study that can offer insights for future research. 

Suggested revisions 

Title and Abstract at line 10: 'L. chinese' change to Liriodendron chinese, maintaining the Italic style;

line 14: 'A. trichopoda' change to Amborella trichopoda;

lines 19, 25 and along the text of the whole manuscript: 'Liriodendron' change to Italic style;

Keywords: 'L. chinese' change to Liriodendron chinese;

lines 54, 65: 'Arabidopsis' change to Italic style;

line 55: 'Fragaria vesca' and 'Branica rapa' change to Italic style;

lines 77, 80: 'Liriodendron' change to Italic style;

lines 94-95: 'L. chinese' change to Italic style;

line 105: 'LcAPP6c' change to Italic style;

Table 1: Please, control the Gene Names and the Locus change to Italic style;

Table S1: Gene Names change to Italic style;

lines 122-123: please control, as the first time mentioning a species, it must be reported in extenso;

line196: 'Arabidopsis thaliana, Oryza sativa, Vitis vinifera, change to Italic style;

Table 2: the name of the species , change to Italics;

line 353: 'Liriodendron' change to Italic style;

line 393: 'Arabidopsis change to Italic style;

lines 460-461, 472, 538-539: 'L. chinese' change to Italic style;

line 469: 'Liriodendron' change to Italic style;

lines 478-479: 'Liriodendron' change to Italic style;

lines 517-518, 529, 531, 601, 640, 651: 'Liriodendron' change to Italic style;

line 650: LcAAAP: change to Italics;

line 653: genes names change to Italics;

Table S7: genes names change to Italic style;

Control all the species names along the text of the manuscript, change to Italic style, and in the References section.

Author Response

Dear editor and reviewers:

I am very grateful to your comments for the manuscript. According with your advice, we amended the relevant part in manuscript.  Since your question mainly focuses on the format, I do not list them all here, but mark all the modifications with a yellow background. All the lines and pages indicated above are in the revised manuscript (Identification, phylogenetic and expression analyses of the AAAP gene family in Liriodendron chinense reveal their putative functions in response to organ and multiple abiotic stresses) and the revisions were marked with a blue background. Thank you again for asking this question, which is of great help to the arrangement of our experimental content and the writing logic of the article. We sincerely hope to receive your comments on our revised content again.

Sincerely yours,

Hu Lingfeng

Round 2

Reviewer 1 Report

After Authors revision Figure 8 is improved, but Figure 5, Figure 10 and 13 are still too small, the quality is still problem;

Please, be carefull, stage of embryo development it is not plant organ, yet;

Despite of it, I would like to underline that Authors improved significantly the manuscript, espacially introduction part and material and methods;

As I mentioned above, some figures need to be improved- authors obtained interesting results and it is strange situation that they do not expose findings duly to share with the readers

Author Response

Dear editor and reviewers1:

I am very grateful to your comments for my manuscript again. According with your advice, we have improved Figures 5, 8, and 10 to convey the relevant information in the article more clearly, and Figure 13 has been split into Figures 13, 14, and 15 to magnify its clarity. Thank you again for asking these question, which is of great help to the arrangement of our experimental content and the writing logic of the article. We sincerely hope to receive your comments on our revised content again.

Sincerely yours,

Hu Lingfeng